# Smart Cancer-Targeting and Super-Sensitive Sensing of Eu^3+^/Tb^3+^-Induced Hyaluronan Characteristic Nano-Micelles with Effective Drug Loading and Release

**DOI:** 10.3390/molecules29215070

**Published:** 2024-10-26

**Authors:** Yupeng Bi, Longlong Li, Jin Liu, Yao Wang, Boying Wang, Yanxin Wang, Christopher D. Snow, Jun Li, Matt J. Kipper, Laurence A. Belfiore, Jianguo Tang

**Affiliations:** 1Institute of Hybrid Materials, National Center of International Joint Research for Hybrid Materials Technology, National Base of International Science & Technology Cooperation on Hybrid Materials, Qingdao University, 308 Ningxia Road, Qingdao 266071, China; b15254156675@163.com (Y.B.); longlonglee@foxmail.com (L.L.); liujin0620@126.com (J.L.); wangyaoqdu@126.com (Y.W.); 13503310534@163.com (B.W.); wangyanxin@qdu.edu.cn (Y.W.); belfiore@engr.colostate.edu (L.A.B.); 2Department of Chemical and Biological Engineering, School of Materials Science and Engineering, School of Biomedical Engineering, Colorado State University, Fort Collins, CO 80523, USA; christopher.snow@colostate.edu; 3School of Materials Science and Engineering, Shanghai University of Engineering Science, Shanghai 201620, China; jacob_lijun@sues.edu.cn

**Keywords:** drug carriers, detection, malignant melanoma, rare earth complexes

## Abstract

To avoid the critical problems of effective drugs not being carried to their targeted cancers and their quantity and location not being sensed in situ, this work presents a completely new innovative strategy to achieve both smart cancer targeting (SCT) and super-sensitive sensing (SSS), where one drug carrier works for effective drug loading and release. Herein, malignant melanoma treatment is used as an example of reliable detection and effective therapy. We report two characteristic dumbbell-like nano-micelles and spherical-like nano-micelles of hyaluronan induced by the Eu^3+^/Tb^3+^ complexes for effective drug loading and release, respectively. These special Eu^3+^/Tb^3+^-loaded nano-micelles (marked as ENM and TNM) have strong and sharp red/green luminescence that can sensitively detect the malignant melanoma drug dacarbazine through changes in fluorescence intensity. Cytotoxicity experiments confirmed that both ENM and TNM are not toxic to normal cells at very high concentrations of 4 mM. However, when loaded with cancer drugs (D-ENM and D-ENM), they both killed cancer cells with more than 40% efficacy at this concentration. The in vivo experiments confirmed that D-ENM and D-TNM can effectively target cancer cells in tissue and effectively impede cancer growth. The detection limits of ENM and TNM in sensing cancer drugs can reach 0.456 μg/mL and 0.139 μg/mL, respectively. Therefore, the reported Eu^3+^/Tb^3+^-induced hyaluronan nano-micelles (ENM and TNM) are distinguished carriers of this cancer drug and excellent in situ sensors, and they have highly therapeutic effects with extremely low toxicity to normal cells.

## 1. Introduction

It is a goal for both smart cancer targeting (SCT) and super-sensitive sensing (SSS) to be achieved with one effective drug carrier that works for both loading and release. The critical problems in realizing these aims are that effective carrier drugs must be able to reach the target cancer, be sensed to determine a quantity and location, and effectively kill cancer cells with extremely low toxicity to normal tissue. Skin cancer is the most common type of cancer in humans, and malignant melanoma is the most aggressive and treatment-resistant form [1]. Melanoma accounts for about 2% of skin cancer cases but is responsible for most skin cancer deaths [2,3]. Although surgical resection is one of the most commonly used methods for treating melanoma, there are disadvantages to this therapy, such as difficulty in completely examining the margin of resection, the high risk of local recurrence, and the invasive nature of the surgery, leading to poor patient compliance [4]. Chemotherapy is a classic cancer treatment strategy and has become a potential treatment modality for anti-tumor activity. However, chemotherapy is not satisfactory for treating malignant melanoma and usually causes multi-organ toxicity owing to systemic administration [5]. The development of new chemotherapeutic drugs to treat tumors is hindered by high costs and long experimental cycles.

A more precise chemotherapeutic delivery could be achieved by delivery vehicles that selectively target malignant cells and locally report drug concentrations, providing precise and real-time feedback. Such drug delivery strategies would enhance the efficacy of malignant tumor treatments by selectively killing tumor cells and offering drug release visualization.

Nanoparticles usually have higher bioavailability and better therapeutic effects than traditional drugs [6]. Nanoparticle-based drug delivery vehicles can locally increase drug concentrations in or near malignant cells while also systemically reducing drug metabolism and elimination [7]. These features can enhance therapeutic efficacy.

Lanthanide fluorescent materials can emit high-brightness optical signals upon excitation with suitable wavelengths of light [8] and may have transformative impacts in biological imaging. For example, in medical imaging, these materials can be used in tumor diagnosis and treatment. Wu et al. developed a tumor-selective lanthanide nanomedicine and achieved efficient tumor fluorescence localization and radionuclide therapy via yttrium-90 labeling [9]. Raphael et al. developed a second-generation Eu^3+^ complex, enabling light-guided surgery and postoperative single-photon luminescence microscopy for solid tumors through Cherenkov radiation, with improved detection sensitivity [10]. Meng et al. prepared europium-based metal–organic frameworks (Eu MOFs) through a simple one-step hydrothermal synthesis method and used them as fluorescent probes to detect ciprofloxacin and aluminum ions (Al^3+^) in aqueous environments, achieving high sensitivity and good selectivity [11]. In general, lanthanide fluorescent materials have promising prospects for use in medical imaging in cancer treatment.

This work aims to develop an intelligent in situ fluorescence-sensing technology for cancer treatment, especially for the challenging treatment of malignant melanoma. We adopted an innovative approach by synthesizing complexes containing Eu^3+^ and Tb^3+^ and binding them with hyaluronan to form nano-micelles. These complexes were chosen for their high fluorescence intensity, clear emission band, long fluorescence lifetime, and insensitivity to photobleaching. However, since these complexes are inherently hydrophobic, toxic, and non-biocompatible, we successfully fabricated our fluorescent nano-micelles by embedding them in naturally hydrophilic hyaluronan, thereby addressing these issues. Moreover, hyaluronan is a ligand for the CD44 receptor, overexpressed by many types of cancer cells. Thus, using hyaluronan to compatibilize the luminescent complexes also endowed the nano-micelles with tumor-selective capabilities.

We verified the tumor selectivity and cytocompatibility of these micelles using cell cytotoxicity tests and tumor model experiments. The results showed that the nano-micelles effectively detected dacarbazine and monitored the drug’s release through changes in fluorescence intensity. In addition, the anti-tumor effects were demonstrated in both in vitro and in vivo models, demonstrating their potential clinical utility (Figure 1).

In summary, this study exploited the beneficial properties of nano-micelles and lanthanide fluorescent materials to develop a nano-micelle-based selective drug delivery system based on lanthanide complexes with a biocompatible polysaccharide for real-time drug content and therapy monitoring in stage III malignant melanoma. These results lay a solid foundation for developing a novel theragnostic nanoparticle family for cancer therapy.

## 2. Results and Discussion

### 2.1. Hyaluronan Nano-Micelles Loaded with Lanthanides (Eu^3+^ and Tb^3+^)

We observed the ENMs and TNMs via TEM to determine their nanostructures. Nano-micelle preparations with the concentrations listed in Table 1 resulted in the structures shown in Figure 2. TEM images of the sample A ENMs revealed an unusual dumbbell-shaped structure. This structure may be beneficial for improving the stability of nano-micelles in vivo because the dumbbell-shaped ends may more easily interact with biomolecules, while the middle part may provide a larger space to accommodate drug molecules [12].

The TEM image of the G TNMs in Figure 2 shows a regular spherical morphology, indicating that at this concentration, the TNMs may undergo a more uniform self-assembly process during synthesis. Spherical nano-micelles generally have better biocompatibility and lower surface energy, which helps to reduce nonspecific interactions with biomolecules and thus reduce potential toxicity.

Figure 3a,b show transmission electron microscopy images of ENMs and TNMs prepared according to conditions A and G in Table 1. Owing to the different concentrations of the ligands and lanthanides, these samples showed different nano-micelle shapes. The Eu^3+^ complex sample (a) showed dumbbell-shaped particles with an average size of 153 nm. In the Tb^3+^ complex sample (b), spherical-shaped micelles were observed. The size distributions of the Eu^3+^ and Tb^3+^ complex samples are shown in Figure 3a,b, respectively. Compared with the europium complex sample, the terbium complex sample had a higher tendency to agglomerate, so the resulting nano-micelles were spherical. Considering the size, stability, and fluorescence intensity of the samples, we selected the Eu^3+^ and Tb^3+^ concentrations in samples A and G for further experiments.

Notably, the nano-micelles in this size range meet the requirements for the enhanced permeability and retention (EPR) effect, where nano-micelles avoid filtration through the liver and spleen and accumulate in tumor tissue.

High-resolution transmission electron microscopy was used to probe the structure of these ENMs and TNMs further and evaluate the elemental distribution of the spherical nano-micelles. Figure 3c–h show the distribution of the Eu, Tb, and Na elements in samples A and G, and Figure 3 shows the distribution of the corresponding elements. Figure 3c,f show the TEM images of the ENMs and TNMs, respectively. The distribution of Eu (red) and Na (blue) in the ENMs is shown in Figure 3d,e. The distribution of Tb (yellow) and Na (orange) in the TNMs is shown in Figure 3g,h. Energy spectrum analysis revealed the distribution of elements in the spherical nano-micelles. The Na element distribution was different from that of the lanthanide elements, as the former only existed in the center of the sphere, while the latter was distributed throughout the entire micelle. This elemental map is consistent with the formation of nano-micelles. More details are summarized in Appendix A.

### 2.2. Photophysical Properties of Hyaluronan Nano-Micelles Loaded with Lanthanides (Eu^3+^ and Tb^3+^)

The photophysical properties of the Eu^3+^ and Tb^3+^ complex fluorescent micelles were tested using a fluorescence spectrophotometer.

Figure 4a,b show the fluorescence excitation and emission spectra of the Eu^3+^ complex and ENMs. The Eu^3+^ complex has a wide absorption band of 360 nm to 420 nm, with a maximum excitation wavelength of 400 nm and its strongest emission wavelength at 620 nm. Five characteristic absorption peaks are in the emission spectra of all complexes in Figure 4b, corresponding to the ^5^D_0_^7^F_0_ (580 nm), ^5^D_0_^7^F_1_ (590 nm), ^5^D_0_^7^F_2_ (612 nm), ^5^D_0_^7^F_3_ (652 nm), and ^5^D_0_^7^F_4_ (704 nm) transitions of Eu^3+^ [13]. Figure 4d,e show the fluorescence excitation and emission spectra of the Tb^3+^ complexes and TNMs. Figure 4d shows that the Tb^3+^ complex has a wide absorption band between 350 and 400 nm. This broad band is caused by the π-π* electronic transition between 4-BBA and phen in the complex. The maximum excitation wavelength of the Tb^3+^ complex is 375 nm. The emission spectra of all complexes in Figure 4e show four characteristic peaks corresponding to the ^5^D_4_^7^F_6_ (490 nm), ^5^D_4_^7^F_5_ (546 nm), ^5^D_4_^7^F_4_ (580 nm), and ^5^D_4_^7^F_3_ (620 nm) transitions of Tb^3+^ [14]. Unlike traditional Eu^3+^ and Tb^3+^ ternary complexes [15,16,17], Eu(DBM)_3_phen and Tb(4-BBA)_3_phen can be excited by blue light with lower energy, which is less harmful to biological tissues. Compared with Eu(DBM)_3_phen and Tb(4-BBA)_3_phen complexes, the fluorescence intensities of the strongest ENM and TNM emission bands were significantly enhanced at 612 nm (Eu) and 546 nm (Tb), respectively. We hypothesize that this enhancement is due to the nano-micelle structure excluding the solvent from the Eu^3+^ and Tb^3+^ complexes, preventing the quenching of the Eu^3+^ and Tb^3+^ complexes in the solution. Alternately, the rigidification of the luminescent complexes in the nano-micelles’ core may reduce non-radiative energy loss.

Fluorescence lifetime and quantum yield are important parameters for characterizing fluorescent nano-micelles. The very long fluorescence lifetimes of lanthanide complexes compared with those of organic fluorophores can also distinguish signals and enhance signals relative to background autofluorescence in biological samples. The fluorescence lifetimes of the ENMs and TNMs are shown in Figure 4c,f: 146 µs (Eu) and 492 µs (Tb), respectively. The quantum yields were 53.8% (Tb) and 62.9% (Eu), respectively. The resulting Eu^3+^ and Tb^3+^ complexes have high fluorescence intensity, wide emission bands, and long fluorescence lifetimes.

### 2.3. Detection of Dacarbazine via Hyaluronan Nano-Micelles Loaded with Lanthanides (Eu^3+^ and Tb^3+^)

Hyaluronan nano-micelles loaded with Eu^3+^ and Tb^3+^ (ENMs and TNMs) have the advantages of water solubility, non-toxicity, strong emissions, and long fluorescence lifetimes.

The feasibility of using these lanthanide-loaded hyaluronan nano-micelles to develop highly sensitive anticancer drug sensors was evaluated. The structures of the ENMs and TNMs loaded with drugs were characterized via TEM. Figure 5a,b show the TEM images of the ENMs and TNMs loaded with DTIC, respectively. After the drug addition, both nano-micelles became larger. The ENMs loaded with DTIC were larger than the TNMs (which again adopted a spherical shape). The zeta potential of the ENMs was −15.7 eV, and the zeta potential after drug loading was −13.2 eV. The zeta potential of the TNMs was −17.3 eV and −14.4 eV after drug loading, consistent with successful drug loading into nano-micelles.

Figure 5c illustrates the drug release curves of the ENMs and TNMs with DTIC at a concentration of 100 µM. These curves were obtained through the in vitro drug release experiments in which drug-loaded micelles were placed in a semipermeable membrane and then in a beaker containing phosphate buffer (PBS). During the experiment, the release solution was changed every 0.5 h, and the drug concentration in each release solution was recorded using a UV spectrophotometer. The drug release curve was obtained by repeating the experiment three times (*n* = 3).

The drug release profiles from both nano-micelle types had no apparent burst, exhibiting a sustained release due to the electrostatic binding of drugs inside the micelles. Sustained release is very important for drug delivery systems because it can simultaneously reduce toxicity and improve drug efficacy. Sustained release can also reduce the frequency of drug administration, thereby improving patient compliance. In this experiment, about 80% of the loaded drug was released from the nano-micelles after about 2.5 h, indicating that the nano-micelles had a sustained release. The drug loading rate was about 17%, as indicated by the UV absorption measurements. More details are summarized in Appendix A.

X-ray photoelectron spectroscopy also confirmed the ability of HA to encapsulate the Eu^3+^ and Tb^3+^ complexes, as well as the ability of ENMs and TNMs to bind to DTIC.

Figure 6a,d show the XPS spectra of the ENMs and TNMs. Figure 6b,e (and Figure 6c,f) show the high-resolution O1s envelope of the nano-micelles and nano-micelles with and without DTIC, respectively. Figure 6b shows that the O1s spectrum of the ENMs has two peaks, corresponding to the C=O (531.18 eV) of the ENM ligand DBM and the C-O (532.18 eV) of HA in the ENMs. By comparing Figure 6b,c, we can see that the binding energies of the C-O and C=O in the ENMs increased by −0.2 eV and −0.25 eV, respectively, after adding HA, indicating that O in the HA complexed with Eu^3+^ and changed the surrounding environment of the Eu^3+^ complex, reducing the binding energies. Thus, HA interacts with Eu^3+^ and the complex to form ENMs. The same is true for TNMs, as shown in Figure 6e, where the O1s spectrum of the TNMs had two peaks corresponding to the C=O (531.38 eV) of the TNM ligand 4-BBA and the C-O (532.18 eV) of HA in the Tb nano-micelles. By comparing Figure 6e,f, we can see that the binding energies of the C-O and C=O also increased in the TNMs. The high-resolution O1s spectrum after DTIC loading (Figure 6f) showed two peaks at 531.48 eV and 532.88 eV. Compared with that of Figure 6e, the binding energy of the O1s in Figure 6f changed, indicating that DTIC interacted with the TNMs.

Fluorescence spectroscopy was used to evaluate the fluorescence signal response of the ENMs and TNMs to DTIC, and their sensitivity to different DTIC concentrations was determined. Figure 7 shows the fluorescence emission spectra of the ENMs and TNMs in the presence of different DTIC concentrations. The fluorescence emission spectra of Eu^3+^ (λ_ex_ = 405 nm) and Tb^3+^ (λ_ex_ = 375 nm) had maxima at λ_em_ = 620 nm and λ_em_ = 546 nm^+^, respectively. Figure 7a,c show that the fluorescence signal intensity of the ENMs and TNMs gradually decreased with increasing DTIC concentration. When the DTIC concentration was 10 µg/mL, the fluorescence intensity decreased by more than 50%. The quenching efficiency of the ENMs and TNMs was further investigated.

The fluorescence quenching in the presence of DITC can be quantified using the linear Stern–Volmer Equation (1) [18] in the 0–200 µg/mL concentration range:(1)$$\frac{F_{0}}{F}=1+K_{sv}\left[Q\right]$$
where *F*_0_ and *F* are the fluorescence intensities of the ENMs and TNMs at 612 nm and 546 nm in the absence or presence of DTIC; *K_sv_* is the Stern–Volmer quenching constant; and [*Q*] is the drug concentration. Figure 7b,d show a strong correlation between the quenching efficiency (*F*_0_/*F*) of the ENMs and TNMs and the DTIC concentration in a range of 0–200 µg/mL (R^2^ = 0.9644, 0.9157). The fitted linear equations are as follows:(2)$$\frac{F_{0}}{F}=8.2356\left[Q\right]+1.0087$$
(3)$$\frac{F_{0}}{F}=67.226\left[Q\right]+0.8969$$

The LOD values of the ENMs and TNMs were 0.456 μg/mL and 0.139 μg/mL, respectively, calculated as $\frac{3\delta }{s}$ (signal-to-noise ratio of 3) in a concentration range from 0 to 200 µg/mL, where δ represents 20 blank measurements. The standard deviation of the values, S, is the slope of the fitted curve.

Fluorescence-based DTIC assays can be confounded by the presence of other drugs. Therefore, to test the ability of the ENMs and TNMs to selectively detect DTIC, we evaluated their fluorescence intensity in the presence of nine other drugs: pemetrexed disodium (SPD), catechin (CT), quercetin (QR), gemcitabine (GEM), vorinostat (SAHA), kaferol (KF), oxaliplatin (OX), fludarabine (FA), imatinib (IM), and dacarbazine (DTIC), all at a concentration of 100 µg/mL. The inhibitory effects of each anticancer drug on ENMs and TNMs under the same test conditions and environment are shown in Figure 7c,f. The relative fluorescence intensity (*F*_0_ − *F*)/*F*_0_ value of DTIC was the lowest.

This indicates that the ENMs and TNMs have a high sensitivity for DTIC.

### 2.4. Cytotoxicity of Hyaluronan Nano-Micelles Loaded with Lanthanides (Eu^3+^ and Tb^3+^)

The CCK8 assay can be used to quantify cell viability. Figure 8 shows a CCK8 cytotoxicity assay for the ENMs (a and b) and TNMs (c and d) on A375 melanoma cells (a and c) and L02 normal hepatocytes (b and d). The ENMs (8 mM) and TNMs (8 mM) reduced the metabolic activity of L02 normal hepatocyte cells by approximately 40%, reaching the cytotoxic threshold. As expected, owing to the selective effect of HA, both nano-micelles were more lethal toward the A375 melanoma cells and virtually non-toxic to the L02 normal hepatocytes at low concentrations; the cytotoxicity of the nano-micelles to the A375 melanoma cells significantly increased after being loaded with DTIC, while the cytotoxicity of DTIC to normal cells was lessened owing to nano-micelle encapsulation.

The transmission electron microscopy images (Figure 9) show that the nano-micelles aggregated in the tumor microenvironment and bound to tumor cells. In both cases, HA reduced the cytotoxicity. These results confirmed our original hypothesis that HA-embedded Eu(DBM)_3_phen and Tb(4-BBA)_3_phen complexes can reduce the cytotoxicity of Eu^3+^ and Tb^3+^ complexes, resulting in good cytocompatibility. The CD44-selective effect of HA can significantly enhance the killing effect of drugs on tumor cells [19,20].

### 2.5. In Vivo Anti-Tumor Evaluation of Hyaluronan Nano-Micelles Loaded with Lanthanides (Eu^3+^ and Tb^3+^)

The anti-tumor activity of the ENMs and TNMs was evaluated in a murine malignant melanoma model.

ENMs, TNMs, ENMs loaded with DTIC, and TNMs loaded with DTIC inhibited tumor growth compared with saline, with limited anticancer effects for free drugs and nano-micelles alone. The poor therapeutic effect of free drugs may be due to the rapid metabolism of free drugs in the body or a lack of effective accumulation. Drug-loaded nano-micelles, especially in the late stage of the experiment, showed the best anti-tumor effects. The final tumor volume for the null treatment was approximately twice that of the drug-loaded nano-micelle treatment (Figure 10b), providing convincing evidence that the drug-loaded nano-micelles can act as drugs in vivo. The safety of the drug was assessed by recording the changes in the body weights of the mice (Figure 10c). The weights of the mice in each group were stable during the treatment, consistent with a favorable physiological safety profile.

Effective drug absorption by cells is pivotal for securing superior therapeutic effects. The chemical stability of nano-micelles, coupled with their distribution and retention within tumor tissues, plays a vital role in ensuring effective drug delivery. Research indicates that nano-micelles between 50 and 200 nm can effectively infiltrate and concentrate within the tumor thanks to their ability to engage with the tumor’s extracellular matrix and the pressure of interstitial fluids. The hydrophilic nature of hyaluronan can augment the stability and fixation of nano-micelles within the tumor, thus elevating their therapeutic efficacy. As a result, we have assessed the prepared co-delivery platform’s cellular uptake, tissue distribution, and fluorescence stability within a tumor following localized injection. Under 405 nm excitation, the Eu^3+^ and Tb^3+^ micelles produced red and green fluorescence, respectively, enabling drug localization and tracking both in cells and in vivo. Immediately after the drug injection (0 h), fluorescence appeared in the tumor tissue area, with high fluorescence intensity and poor fluorescence dispersion in the tumor tissue (Figure 11). After 0.5 h of drug injection, the fluorescence intensity increased, consistent with carrier fluorescence increasing concomitant with drug release. At 1 h and 1.5 h after drug injection, we observed carrier aggregation in the form of punctate bright spots. At 2 and 2.5 h, the fluorescence was evenly distributed in the tissues, consistent with the carriers having dissolved (Figure 11). Owing to the complex internal environment, the nano-micelles were present in the tumor tissue as aggregates. Transmission electron micrographs confirmed that the hyaluronan-containing nano-micelles associated with tumor cells, likely through a CD44-hyaluronan receptor–ligand interaction [21].

## 3. Experiments

### 3.1. Materials

Europium chloride hexahydrate (EuCl_3_·6H_2_O) and terbium chloride hexahydrate (TbCl_3_·6H_2_O) were purchased from Shandong Desen Lanthanide Co., Ltd. (Shandong, China) Dibenzoyl methane (DBM), 4-benzoyl benzoic acid (4-BBA), 1,10-phenanthroline (phen), and sodium hyaluronate (HA) were purchased from Shanghai McLean Biochemical Co., Ltd. (Shanghai, China). Dacarbazine (DTIC) was purchased from Aladdin Industrial Co., Shanghai, China. Other chemical reagents were purchased from Shanghai Chemical Reagent Co., Ltd. (Shanghai, China). BPS buffer (pH 6.2–6.4) was purchased from Shanghai McLean Biochemical Co., Ltd. (Shanghai, China). Deionized water was used to prepare all aqueous solutions. The above products are of analytical grade and were used without further purification.

### 3.2. Preparation of Hyaluronan Nano-Micelles Loaded with Lanthanides (Eu^3+^ and Tb^3+^)

The Eu^3+^ and Tb^3+^ complexes were synthesized as follows. EuCl_3_·6H_2_O, TbCl_3_·6H_2_O, DBM, 4-BBA, and 1,10-phenanthroline (phen) were dissolved in absolute ethanol, and HA was dissolved in deionized water. In total, 1 mL each of EuCl_3_·6H_2_O and DBM (or TbCl_3_ and 4-BBA) at the concentrations shown in Table 1 was mixed and stirred for 30 min at room temperature. The pH of the mixture was then adjusted to 7–8 with 1 mol/L NH_4_·OH. After neutralization, 1 mL of the solution was added, and the mixture was stirred continuously for 2 h. The resulting Eu^3+^ or Tb^3+^ complex was colorless, transparent, and soluble in ethanol. In total, 4 mL of the HA solution was then added to the lanthanide complex solution, and the solution was stirred for 1 h, followed by centrifugation at 10,000 rpm for 10 min. Then, the supernatant was removed, and ethanol was added to form the HA lanthanide complex nano-micelles.

**Table 1 molecules-29-05070-t001:** Concentrations used to prepare the Eu^3+^ and Tb^3+^ complex hyaluronan nano-micelles.

Samples	C_Eu_	C_Tb_	C_DBM_	C_4-BBA_	C_phen_	C_HA_
	(mol/L)	(mol/L)	(mol/L)	(mol/L)	(mol/L)	(g/mL)
A	1.0 × 10^−3^	0	3.0 × 10^−3^	0	1.0 × 10^−3^	1.0 × 10^−3^
B	2.0 × 10^−3^	0	6.0 × 10^−3^	0	3.0 × 10^−3^	1.0 × 10^−3^
C	1.0 × 10^−2^	0	3.0 × 10^−2^	0	1.0 × 10^−2^	0.8 × 10^−3^
D	2.0 × 10^−2^	0	6.0 × 10^−2^	0	3.0 × 10^−2^	0.8 × 10^−3^
E	0	1.0 × 10^−3^	0	3.0 × 10^−3^	1.0 × 10^−3^	1.0 × 10^−3^
F	0	2.0 × 10^−3^	0	6.0 × 10^−3^	3.0 × 10^−3^	1.0 × 10^−3^
G	0	1.0 × 10^−2^	0	3.0 × 10^−2^	1.0 × 10^−2^	0.8 × 10^−3^
H	0	2.0 × 10^−2^	0	6.0 × 10^−2^	3.0 × 10^−2^	0.8 × 10^−3^

### 3.3. Fluorescence Characteristics of Hyaluronan Nano-Micelles Loaded with Lanthanides (Eu^3+^ and Tb^3+^)

The Eu(DBM)_3_phen had an excitation wavelength of 405 nm, an emission wavelength of 620 nm, and an emission slit width of 1 nm. For Tb(4-BBA)_3_phen, an excitation wavelength of 374 nm, an emission wavelength of 548 nm, and an emission slit width of 1 nm were used. The fluorescence lifetime and quantum yield of the samples were determined.

### 3.4. Characterization of Hyaluronan Nano-Micelles Loaded with Lanthanides (Eu^3+^ and Tb^3+^)

The morphology and elemental map of the Eu^3+^ and Tb^3+^ nano-micelles were obtained using a JEM-2100F (JEOL Ltd., Tokyo, Japan) transmission electron microscope (TEM).

Each sample was sonicated for 5–6 min. The sonicated solution was then dropped onto a copper grid and evaporated for 1 or 2 min at room temperature. The photoluminescence spectra of the samples were obtained using a photoluminescence spectrometer (FLS1000, Edinburgh, UK), and the fluorescence lifetime and quantum yield of the Eu^3+^ and Tb^3+^ nano-micelles were evaluated. X-ray photoelectron spectroscopy (XPS) was performed using a Thermo Scientific K-Alpha instrument (Thermo Scientific, Waltham, MA, USA).

### 3.5. Sensing Anticancer Drugs with Hyaluronan Nano-Micelles Loaded with Lanthanides (Eu^3+^ and Tb^3+^)

#### 3.5.1. Detection of Dacarbazine (DTIC)

DTIC solutions were prepared at concentrations ranging from 0.01 mg/mL to 0.2 mg/mL. From these DTIC solutions, 1 mL was added to 3 mL of a 1 mM Eu^3+^ nano-micelle solution and 3 mL of a 10 mM Tb^3+^ nano-micelle solution and stirred for 0.5 h at room temperature in the dark. Then, the resulting samples were subjected to fluorometric spectrophotometry.

The Eu^3+^ was characterized by an excitation wavelength of 400 nm, an emission wavelength of 620 nm, and an emission slit width of 0.5 nm.

The Tb^3+^ was characterized by an excitation wavelength of 372 nm, an emission wavelength of 546 nm, and an emission slit width of 1 nm. The limit of detection (LOD) of DTIC in the Eu^3+^ and Tb^3+^ nano-micelles was calculated. All samples were tested in triplicate (*n* = 3) at room temperature with the same parameters.

#### 3.5.2. Sensitivity of Sodium Hyaluronate Nano-Micelles Loaded with Lanthanides (Eu^3+^ and Tb^3+^) to Anticancer Drugs

Ten drug aqueous solutions of pemetrexed disodium (SPD), catechin (CT), quercetin (QR), gemcitabine (GEM), vorinostat (SAHA), kaempferol (KF), oxaliplatin (OX), fludarabine (FA), imatinib (IM), and dacarbazine (DTIC) were prepared at a concentration of 1 mg/mL. Then, 1 mL of the drug solution was added to 3 mL of a 1 mM europium micelle solution and 3 mL of a 10 mM terbium micelle solution; the mixtures were then stirred at 8000 rpm for 0.5 h at room temperature and sonicated for 0.5 h. The supernatant was removed via centrifugation, and dacarbazine-loaded nano-micelles were obtained via rinsing with deionized water. Spectrophotometry was performed as described above.

### 3.6. In Vitro Release

Drug-loaded ENMs and TNMs (4 mL, 10 mM) were dissolved in 10 mL of PBS and dialyzed with PBS at 37 °C. The external solution was changed periodically; 5 mL of the external solution was extracted each time for quantitative analysis via UV absorption, and the same volume of fresh PBS was added. Drug release was measured via UV spectrophotometry with a UV-7558 instrument (Shanghai Metash Instruments Co., Ltd., Shanghai, China). The samples were dissolved in 10 mL of PBS and dialyzed with PBS at 37 °C. The external solution was changed periodically, and 5 mL was extracted each time for quantitative analysis via UV absorption. The same volume of fresh PBS was then added. Drug release was tested via UV spectrophotometry using a UV-7558 spectrophotometer.

### 3.7. Cell Culture and Animal Care and Use

Human malignant melanoma (A375) and human normal liver cells (L02) were selected to study their micelle cytotoxicity and cellular uptake. Cells were cultured according to standard procedures. BALB/c nude mice (15–18 g; 7 weeks old; female) were purchased from SPF (Beijing) Biotechnology Co., Ltd., Beijing, China. All animal experiments were reviewed by the Laboratory Animal Management and Ethics Committee of Qingdao University (QDU-AEC-2021197). Animal studies were conducted according to the Guide for the Care and Use of Laboratory Animals of the National Research Council.

### 3.8. Biocompatibility Testing

ENM and TNM cytotoxicity were determined using a standard CCK8 assay protocol with A375 cells (human hepatic malignant melanoma cells) and L02 (human normal liver cells). A375 and L02 cells in the logarithmic growth phase were counted, and the cell concentration was adjusted. A375 and L02 cells were seeded into a 96-well plate at 6 × 10^3^/well and 8 × 10^3^/well, respectively. The cells were divided into 5 groups: the control group and groups 1 through 4. The control group was treated with 100 μL/well with a complete medium. Groups 1 through 4 were treated with a 100 μL/well nano-micelle working solution at concentrations of 2 mM, 4 mM, 6 mM, 8 mM, 10 mM, 12 mM, 14 mM, and 16 mM with ENMs, TNMs, and ENMs loaded with DTIC, or TNMs loaded with DTIC. Each treatment was performed on triplicate wells, and the cells were then incubated for another 24 h. After 24 h, the medium containing the experimental or control treatment was removed, and the cells were washed with the cell culture medium 3 times. Each well was treated with 100 µL of the medium containing 0.5 mg/mL of the CCK8 reagent, and the cells were then incubated (5% CO_2_, 37 °C) for 4 h. After shaking for 10 min, the absorbance at 570 nm was measured [22,23].

### 3.9. Establishment of the Tumor Model

A mouse model of malignant melanoma was established by subcutaneously injecting A375 cells (2 × 10^6^ cells per mouse) into the left axilla. The anti-tumor study was performed when the tumor volume reached about 80–100 mm^3^.

### 3.10. In Vivo Anti-Tumor Evaluation

Eighteen mice were randomly divided into 6 groups (*n* = 3 per group) and treated with (i) ENMs, (ii) DTIC-loaded ENMs, (iii) TNMs, (iv) DTIC-loaded TNMs, (v) free DTIC, or (vi) normal saline for 10 consecutive days. Every day, 50 μL equal doses of different preparations were injected into the tumor. The body weight was recorded, and the tumor volume was calculated using the formula V = (L × W^2^)/2, where L and W are the longest and shortest diameters. On day 10, the mice were sacrificed, and the weight of the resected tumor was measured and photographed.

The tumors were collected for histological evaluation.

### 3.11. Histopathological Examination

The tumors were excised from the mice for histological examination, and the fluorescence was observed via excitation with 405 nm excitation light. Confocal microscopy (CLSM) was performed using a Zeiss LSM 880 instrument (Carl Zeiss AG, Oberkochen, Germany).

### 3.12. Statistical Analysis

All experiments were performed at least three times, and the results are presented as the mean ± standard deviation. A one-way analysis of variance (ANOVA) was used, and the Tukey test was used to compare the means. The results were evaluated to determine their significance level.

## 4. Conclusions

We successfully synthesized and characterized a novel type of hyaluronan nano-micelles doped with luminescent europium (Eu^3+^) (ENMs) and terbium (Tb^3+^) complexes (TNMs). These nano-micelles not only demonstrated excellent water solubility and cytocompatibility but also had the potential to specifically bind to tumor cells that overexpress hyaluronan receptors. The unique morphological structures of ENMs and TNMs with dumbbell and spherical shapes contributed to their stability and biocompatibility in the body, respectively. The photophysical properties of the strong and sharp fluorescence and long fluorescence lifetimes of both the ENMs and TNMs provided ideal characteristics for biological imaging and drug sensing. In this study, we innovated using novel and special properties, as follows:

Both ENMs and TNMs not only demonstrated excellent water solubility and cytocompatibility but also had the potential to specifically target tumor cell surfaces. The exceptional sensitivity of both ENMs and TNMs in detecting DTIC indicated limits as low as 0.456 μg/mL and 0.139 μg/mL for ENMs and TNMs, respectively, highlighting their high sensitivity for in situ sensing.

We showed that these novel nano-micelles can effectively bind targets and release the chemotherapeutic drug dacarbazine, with drug release following zero-order kinetics for the first 2.5 h, during which more than 80% of the loaded drug is released. Moreover, the nano-micelles sensitively and selectively monitored the dacarbazine concentrations through fluorescence quenching.

Our cytotoxicity assessments (Figure 8) indicated that the nano-micelles had a significant cytotoxic effect on A375 melanoma cells while maintaining low toxicity to L02 normal human liver cells, underscoring the role of hyaluronan in enhancing the tumor-targeting ability of these nano-micelles. The anti-tumor effect of the nano-micelles was significantly enhanced after drug loading. In animal models (Figure 9), we observed fluorescence changes in the nano-micelles in tumor tissues, confirming their potential for significant therapeutic efficacy and safety in cancer treatment.

In summary, the results of our study indicated that these novel hyaluronan nano-micelles have immense potential in smart cancer-targeted therapy and drug release monitoring and sensing, offering an innovative and promising new strategy for cancer treatment. These findings not only lay the groundwork for future cancer therapy research but also provide important insights into developing new theragnostic nanoplatforms.

## Figures and Tables

**Figure 1 molecules-29-05070-f001:**
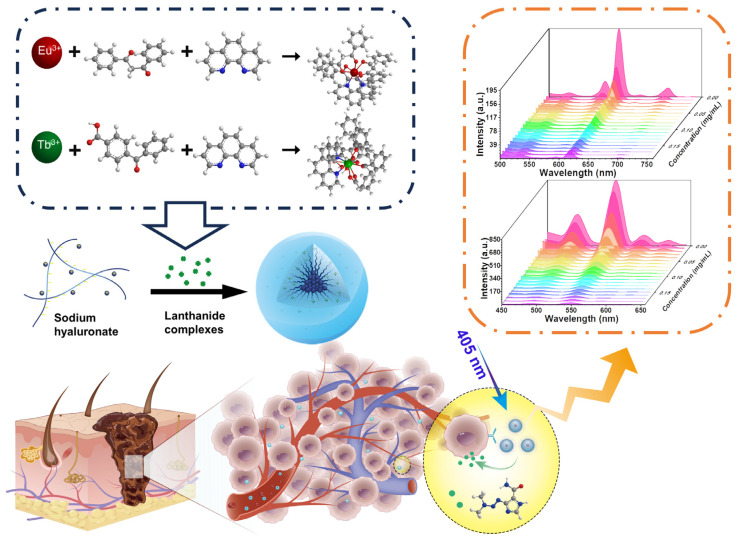
Preparation, chemical structure, and anticancer mechanism of lanthanide hyaluronan nano-micelles [1].

**Figure 2 molecules-29-05070-f002:**
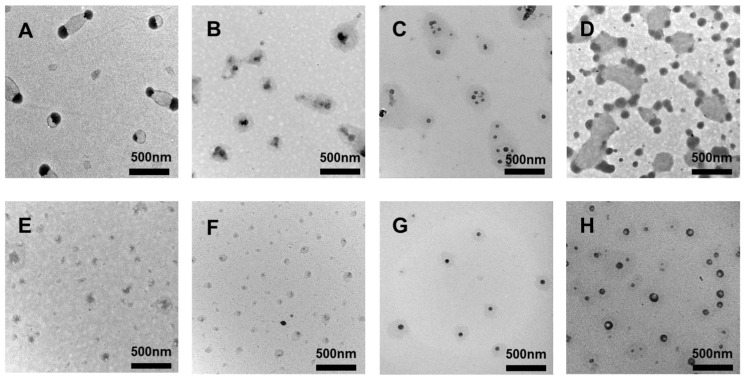
TEM images of samples (**A**–**H**), prepared as per the details in Table 1.

**Figure 3 molecules-29-05070-f003:**
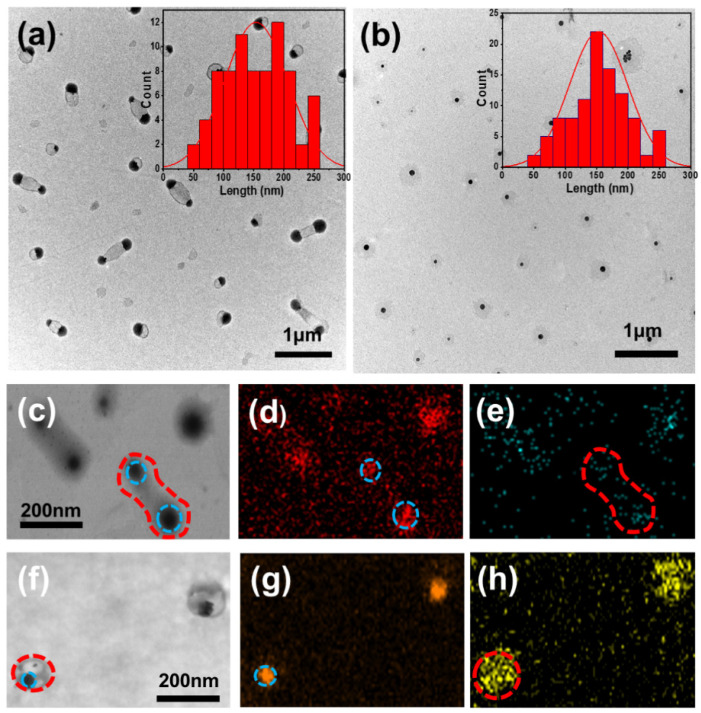
(**a**) TEM image and size distribution of ENMs (sample A). (**b**) TEM image and size distribution of TNMs (sample G). (**c**) TEM image of ENMs. (**d**) Distribution of sodium in ENMs. (**e**) Distribution of europium in ENMs. (**f**) TEM image of TNMs. (**g**) Sodium element distribution in TNMs. (**h**) Terbium element distribution in TNMs.

**Figure 4 molecules-29-05070-f004:**
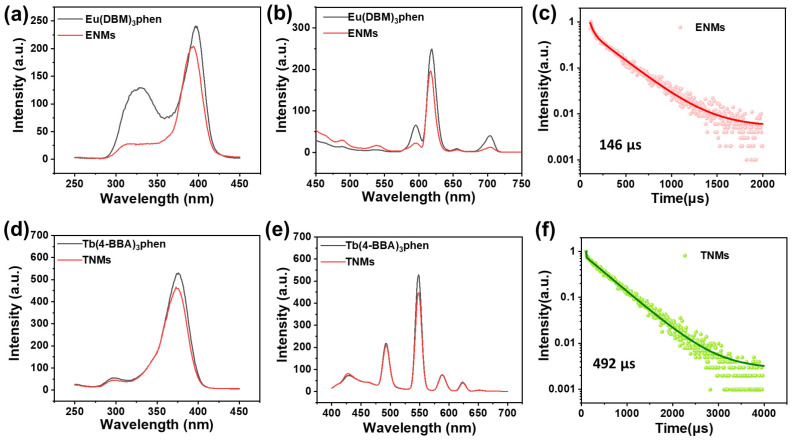
(**a**) Excitation curves of europium complexes and ENMs. (**b**) Emission curves of europium complexes and ENMs. (**c**) Fluorescence lifetimes of ENMs. (**d**) Excitation curves of terbium complexes and TNMs. (**e**) Emission curves of terbium complexes and TNMs. (**f**) Fluorescence lifetimes of TNMs.

**Figure 5 molecules-29-05070-f005:**
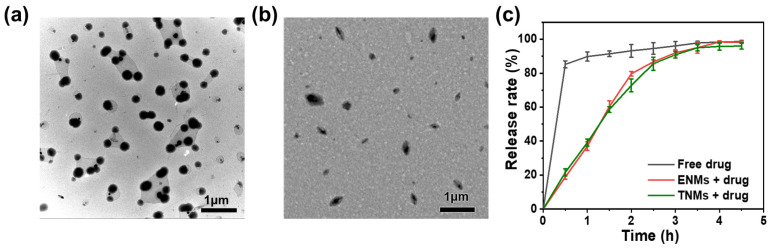
(**a**) TEM images of ENMs loaded with DTIC. (**b**) TEM images of TNMs loaded with DTIC. (**c**) Drug release curves.

**Figure 6 molecules-29-05070-f006:**
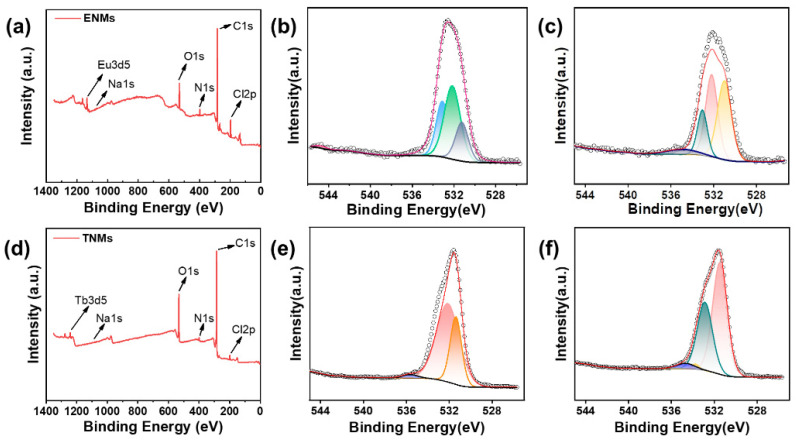
(**a**) ENM XPS spectrum. (**b**) ENM XPS oxygen high-resolution spectrum without drug loading. (**c**) ENM XPS oxygen high-resolution spectrum with drug loading. (**d**) TNM XPS survey spectrum with drug loading. (**e**) TNM XPS oxygen high-resolution spectrum without drug loading. (**f**) TNM XPS oxygen high-resolution spectrum with drug loading.

**Figure 7 molecules-29-05070-f007:**
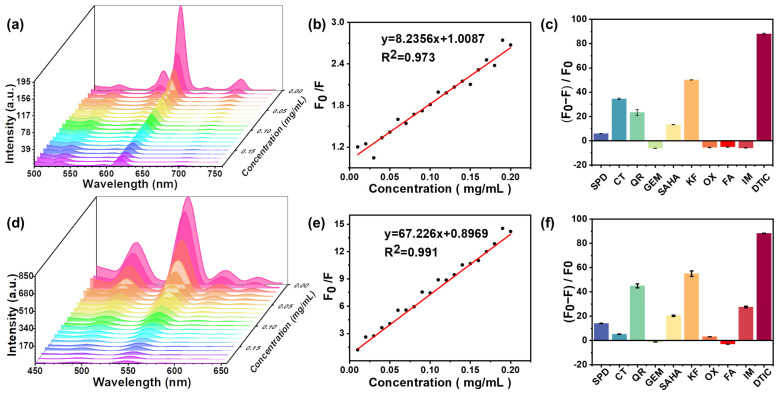
(**a**) Emission images and quenching curves of ENMs quenched with different concentrations of DTIC. (**b**) Curves of ENMs quenched by DTIC. (**c**) Degree of ENM quenching with different drugs at 100 µg/mL. (**d**) Emission images and quenching curves of TNMs quenched with different concentrations of DTIC. (**e**) Curves of TNMs quenched with DTIC. (**f**) Degree of TNM quenching with different drugs at 100 µg/mL. Error bars in e and f indicate a standard deviation of *n* = 3 replicates for each condition.

**Figure 8 molecules-29-05070-f008:**
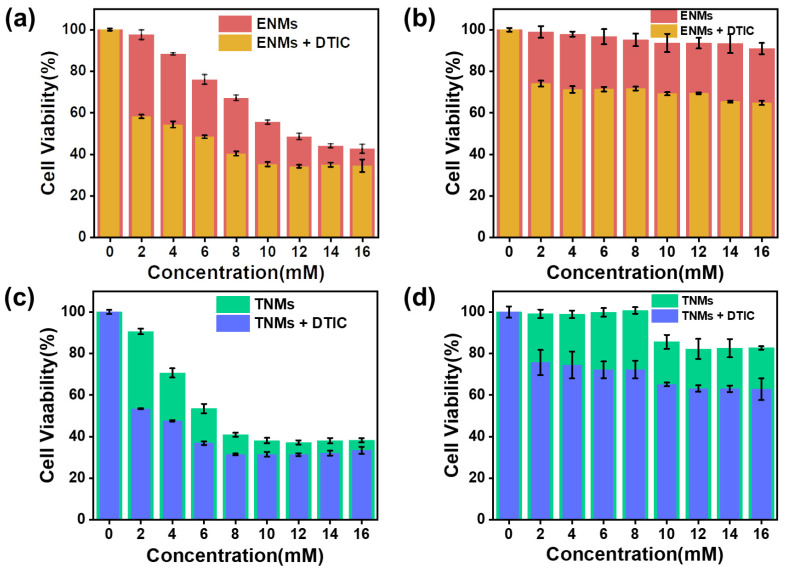
(**a**) Cytotoxicity of unloaded ENMs and loaded ENMs to A375 cells. (**b**) Cytotoxicity of unloaded ENMs and loaded ENMs to L02 cells. (**c**) Cytotoxicity of unloaded TNMs and loaded TNMs to A375 cells. (**d**) Cytotoxicity of unloaded TNMs and loaded TNMs to L02 cells.

**Figure 9 molecules-29-05070-f009:**
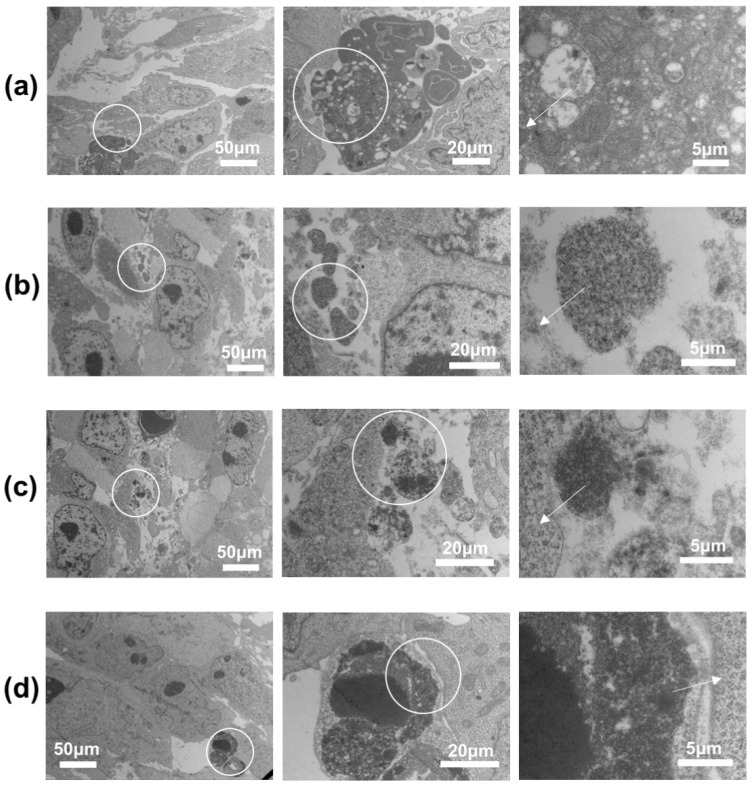
Transmission electron microscope images of tumor tissue after injection with drug-loaded nano-micelles. The arrows in the diagram indicate that the aggregates are encapsulated around the cells. (**a**) Malignant melanoma cells engulf ENMs. (**b**) Malignant melanoma cells engulf TNMs. (**c**) Malignant melanoma cells engulf DTIC-loaded ENMs. (**d**) Malignant melanoma cells engulf DTIC-loaded TNMs.

**Figure 10 molecules-29-05070-f010:**
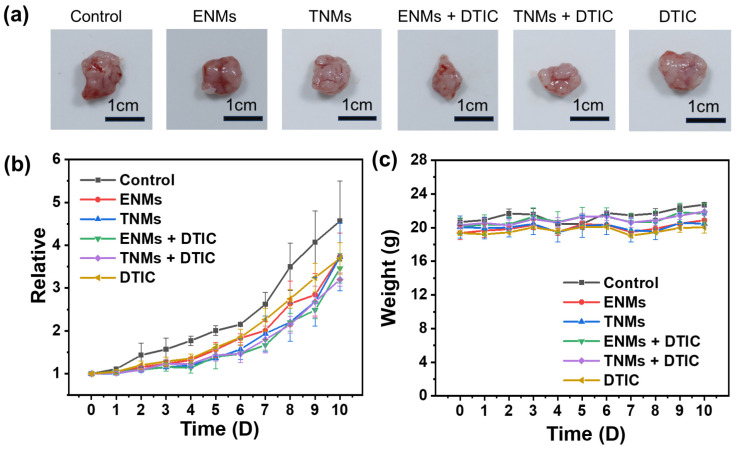
In the malignant melanoma animal model, continuous intratumoral injections of PBS buffer, ENMs, TNMs, ENMs loaded with DTIC, TNMs loaded with DTIC, and free drugs were administered for 10 days. (**a**) Photos of tumors; (**b**) changes in tumor volume; (**c**) changes in body weight of mice. More details are summarized in Appendix A.

**Figure 11 molecules-29-05070-f011:**
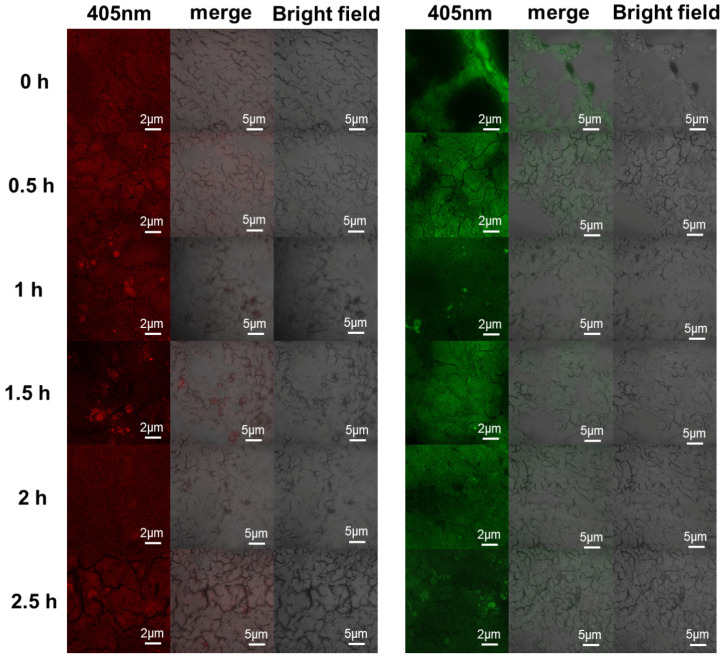
In an animal model of malignant melanoma, after injecting DTIC-loaded ENMs (**left**) and DTIC-loaded TNMs (**right**) into tumor tissue, tumor tissue was observed via fluorescence confocal microscopy at 0 h, 0.5 h, 1 h, 1.5 h, 2 h, and 2.5 h under excitation light at a 405 nm wavelength.

## Data Availability

The data that support the findings of this study are available from the corresponding author upon reasonable request. However, restrictions apply to the availability of these data, which were used under license for this study and are therefore not publicly available.

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
