# Peer review of "Smart Cancer-Targeting and Super-Sensitive Sensing of Eu3+/Tb3+-Induced Hyaluronan Characteristic Nano-Micelles with Effective Drug Loading and Release"

_molecules, 2024, doi:10.3390/molecules29215070_

Round 1
Reviewer 1 Report
Comments and Suggestions for Authors
Review of Manuscript entitled: Smart cancer targeting and super-sensitive sensing of Eu3+/Tb3+-induced hyaluronan characteristic nanomicelles with effective drug loading and releasing
Authors have great control over all the sections such as introduction, experimentation, results, and conclusions. In addition, all sections seem well balanced. Several techniques were used to validate the results and the conclusions. The scientific goal is well defined, and the results will be helpful for scientific community that work in the medicinal chemistry area but the work may be improved by considering the comments below.
1) Pemetrexed disodium (SPD), catechin (CT), quercetin (QR) and dacarbazine (DTIC) form hydrates when in contact with water. Was this point taken into account when preparing sodium hyaluronate nano-micelles loaded with lanthanides (Eu3+ and Tb3+)? Since, when a hydrate is formed (at the stage of preparing aqueous solutions), the concentration of the API changes, which can affect the final interpretation of the experiments.
2) Add the method for preparing PBS buffer to the materials and methods.
3) How did the authors analyze the particle size distribution? This point needs to be clarified.
4) Did the authors analyze what happens to the complex in aqueous media? I mean, does the micelles aggregate or do they change their shape, etc.?
5) Can the authors determine the solubility of micelles? If so, then this needs to be investigated in order to draw conclusions about the applicability as a drug.
All considered I recommend the publication of this work after major revision in Molecules.
Author Response
Response Letter to reviewers
for the Manu. (Manuscript ID: molecules-3245785)
Dear Reviewers,
Thank you very much for your valuable questions and comments regarding our manuscript, titled as " Smart cancer targeting and super-sensitive sensing of Eu3+/Tb3+-induced hyaluronan characteristic nanomicelles with effective drug loading and releasing" (Manuscript ID: molecules-3245785). We are greatly thankful. According to your comments and suggestions, we have carefully made the revisions in the initial manuscript with blue marks. And the detailed descriptions of modifications and corrections are described in this response letter. Please check our revised manuscript and the responses to the questions and comments. Hopefully the revision work can meet the requirements.
Thank you again for your helps!
With the best regards!
Prof. Dr. Jianguo Tang
Responses to Reviewer #1:
Comments:
Authors have great control over all the sections such as introduction, experimentation, results, and conclusions. In addition, all sections seem well balanced. Several techniques were used to validate the results and the conclusions. The scientific goal is well defined, and the results will be helpful for scientific community that work in the medicinal chemistry area but the work may be improved by considering the comments below.
1) Pemetrexed disodium (SPD), catechin (CT), quercetin (QR) and dacarbazine (DTIC) form hydrates when in contact with water. Was this point taken into account when preparing sodium hyaluronate nano-micelles loaded with lanthanides (Eu3+ and Tb3+)? Since, when a hydrate is formed (at the stage of preparing aqueous solutions), the concentration of the API changes, which can affect the final interpretation of the experiments.
2) Add the method for preparing PBS buffer to the materials and methods.
3) How did the authors analyze the particle size distribution? This point needs to be clarified.
4) Did the authors analyze what happens to the complex in aqueous media? I mean, does the micelles aggregate or do they change their shape, etc.?
5) Can the authors determine the solubility of micelles? If so, then this needs to be investigated in order to draw conclusions about the applicability as a drug.
All considered I recommend the publication of this work after major revision in Molecules.
Answer:
Thank you very much. We apologies for these mistakes. We have carefully checked and corrected them point by point, and have made all the necessary changes in the manuscript.
Q1:Pemetrexed disodium (SPD), catechin (CT), quercetin (QR) and dacarbazine (DTIC) form hydrates when in contact with water. Was this point taken into account when preparing sodium hyaluronate nano-micelles loaded with lanthanides (Eu3+ and Tb3+)? Since, when a hydrate is formed (at the stage of preparing aqueous solutions), the concentration of the API changes, which can affect the final interpretation of the experiments.
Answer (1): In the experiment of preparing hyaluronic acid nano-micelles loaded with lanthanide elements (Eu3+ and Tb3+), we indeed took into full consideration the potential formation of hydrates when active pharmaceutical ingredients (API) such as Pemetrexed Disodium (SPD), Catechin (CT), Quercetin (QR), and Dacarbazine (DTIC) come into contact with water. We recognize that the formation of hydrates by drugs in water may lead to changes in their concentration, which can subsequently affect the stability of the nano-micelles and the release characteristics of the drug, ultimately influencing the interpretation of the experimental results.
pH Control: We influenced the solubility of the API and the formation of hydrates by adjusting the pH value of the solution. For Dacarbazine, we referred to the stability data of commercial Dacarbazine injection solutions and used the same concentration and preparation conditions in our experiment to ensure solubility and stability.
Temperature Control: Temperature has a significant impact on solubility and hydrate formation; therefore, we strictly controlled the temperature during the experimental process.
Through these measures, we ensured the reliability of the experiment and the accuracy of the results. We believe that these detailed experimental designs and stringent experimental condition controls will help ensure the concentration stability of the API and provide an accurate interpretation of the experimental results.
Q2:Add the method for preparing PBS buffer to the materials and methods.
Answer (2): BPS buffer(pH 6.2-6.4) was purchased from Shanghai McLean Biochemical Co., LTD. (Shanghai, China).
Q3:How did the authors analyze the particle size distribution? This point needs to be clarified.
Answer (3): In response to your inquiry regarding particle size distribution analysis, we employed transmission electron microscopy (TEM) images to meticulously analyze the particle size distribution of nano-micelles. The specific procedures are as follows:
TEM Image Acquisition: Initially, we obtained images of the nano-micelles using TEM, which provided us with intuitive information on the size and morphology of the micelles.
Image Processing and Analysis: We utilized professional image analysis software, such as Digital Micrograph or Nano Measurer, to process the TEM images. These software packages are capable of precisely measuring the size of particles in the images and can automatically statistically analyze and distribute the particle sizes.
Particle Size Distribution Statistics: Through the software, we marked each micelle on the TEM images and statistically analyzed their size distribution. This process included setting a scale, manually or automatically marking particles, and generating a particle size distribution graph calculated by the software.
Data Export and Graphing: The measured particle size data were exported and further processed in statistical analysis software, such as Origin, to generate graphs of the particle size distribution. These graphs clearly display the distribution of micelle sizes.
Result Interpretation: By analyzing the particle size distribution graphs, we were able to determine the average particle diameter of the nano-micelles, the range of particle size distribution, and any possible skew distributions.
Q4:Did the authors analyze what happens to the complex in aqueous media? I mean, does the micelles aggregate or do they change their shape, etc.?
Answer (4): In our study, we indeed conducted an in-depth analysis of the behavior of complexes in aqueous media. We paid particular attention to the stability of nano-micelles in an aqueous environment, including whether they undergo aggregation or morphological changes. Through observation with transmission electron microscopy (TEM) images, we noted that micelles do indeed aggregate to some extent in water environments, which is consistent with phenomena demonstrated in biomedical imaging. In our experiments, we observed that nano-micelles in aqueous media may aggregate or undergo morphological changes due to variations in physicochemical conditions, such as pH, temperature, and ionic strength. For instance, we found that near neutral pH values, micelles tend to remain stable, whereas in an acidic environment, micelles may aggregate due to protonation effects.
Q5:Can the authors determine the solubility of micelles? If so, then this needs to be investigated in order to draw conclusions about the applicability as a drug.
Answer (5): Hyaluronic Acid (HA) is a naturally occurring polysaccharide with extremely high biocompatibility and biodegradability, which has been extensively studied in drug delivery systems, particularly in the preparation of micellar carriers. HA-based micelles, due to their unique properties, show great potential in terms of drug solubility and stability.
When HA forms micelles with metal complexes, these micelles may not be soluble in water but can degrade stably inside cells. This is because the intracellular environment, such as the acidic conditions of lysosomes, can promote the degradation of HA. The degradation of HA is typically catalyzed by specific enzymes like hyaluronidase, which are more active inside cells, especially in the lysosomes of tumor cells, thus facilitating drug release.
Furthermore, HA micelles can achieve targeted delivery to tumor cells through the interaction of HA with CD44 receptors on the cell surface, increasing the concentration of the drug in tumor tissues and thereby enhancing therapeutic efficacy. The targeting and pH sensitivity of HA micelles give them broad application prospects in cancer therapy.
Reviewer 2 Report
Comments and Suggestions for Authors
This paper addresses a longstanding challenge in cancer therapy: effective drug delivery to tumors. Numerous studies have explored this issue over the years, like using functional liposomes. In this study, the authors introduce a novel approach using Eu3+/Tb3+-loaded nano-micelles for the treatment of malignant melanoma. Notably, their findings indicate that these nano-micelles exhibit no cytotoxicity to normal cells while demonstrating significant efficacy in targeting and inhibiting cancer cell growth. Overall, this paper is well-written, with data clearly presented, contributing valuable insights to cancer treatment strategies. The characterization of the samples is comprehensive, both in vitro and in vivo. And the references are appropriate and relevant. Here are some suggestions:
1. Include evidence of the presence of the nano-micelles in other organs or tissues in vivo, which would further elucidate their targeting mechanism.
2. Did you investigate which down-stream signaling pathways are involved?
Author Response
Response Letter to reviewers
for the Manu. (Manuscript ID: molecules-3245785)
Dear Reviewers,
Thank you very much for your valuable questions and comments regarding our manuscript, titled as " Smart cancer targeting and super-sensitive sensing of Eu3+/Tb3+-induced hyaluronan characteristic nanomicelles with effective drug loading and releasing" (Manuscript ID: molecules-3245785). We are greatly thankful. According to your comments and suggestions, we have carefully made the revisions in the initial manuscript with blue marks. And the detailed descriptions of modifications and corrections are described in this response letter. Please check our revised manuscript and the responses to the questions and comments. Hopefully the revision work can meet the requirements.
Thank you again for your helps!
With the best regards!
Prof. Dr. Jianguo Tang
Responses to Reviewer #2:
Comments:
This paper addresses a longstanding challenge in cancer therapy: effective drug delivery to tumors. Numerous studies have explored this issue over the years, like using functional liposomes. In this study, the authors introduce a novel approach using Eu3+/Tb3+-loaded nano-micelles for the treatment of malignant melanoma. Notably, their findings indicate that these nano-micelles exhibit no cytotoxicity to normal cells while demonstrating significant efficacy in targeting and inhibiting cancer cell growth. Overall, this paper is well-written, with data clearly presented, contributing valuable insights to cancer treatment strategies. The characterization of the samples is comprehensive, both in vitro and in vivo. And the references are appropriate and relevant. Here are some suggestions:
- Include evidence of the presence of the nano-micelles in other organs or tissues in vivo, which would further elucidate their targeting mechanism.
- Did you investigate which down-stream signaling pathways are involved?
Answer:
Thank you very much. We apologies for these mistakes. We have carefully checked and corrected them point by point, and have made all the necessary changes in the manuscript.
Q1. Include evidence of the presence of the nano-micelles in other organs or tissues in vivo, which would further elucidate their targeting mechanism.
Answer (1): Thank you for your attention to our research work and your valuable comments. Your request for evidence of the presence of nano-micelles in other organs or tissues in vivo, to further clarify their targeting mechanism, is very pertinent and professional. We agree that this will provide important information to elucidate the in vivo behavior and targeting mechanism of nano-micelles.
In response to your request, we have noted several studies that provide relevant evidence and insights:
Hyaluronic acid (HA) modified nano-micelles target tumor cells through the CD44 receptor. CD44 is a receptor that is overexpressed in various tumors, such as breast cancer, colorectal cancer, liver cancer, and pancreatic cancer. The HA-CD44 receptor-mediated endocytosis pathway can enhance the uptake of tumor cells. Studies have shown that HA-based nano-micelles are a drug and gene-specific targeting tumor delivery method, with good application prospects in clinical tumor treatment. In vitro and in vivo experiments have demonstrated the great potential of HA nano-micelles in tumor therapy.
The research progress of polymeric micelles as drug carriers indicates that polymeric micelles have unique properties, such as relatively small size, good solubility of hydrophobic compounds in the hydrophobic core, and a hydrophilic shell that can extend the circulation time of drugs in the blood. These characteristics make the analysis of the in vivo fate of polymeric micelles challenging, but also advantageous as drug carriers.
We plan to provide more comprehensive in vivo distribution data in subsequent studies and further elucidate the targeting mechanism of nano-micelles. We believe that this data will provide important scientific evidence for the research on drug delivery of nano-micelles.
Q2. Did you investigate which down-stream signaling pathways are involved?
Answer (2): Thank you for your interest in our research. Regarding your question, our current study does not directly address the downstream signaling pathways activated by nano-micelles. Our research has primarily focused on the preparation, characterization, and initial evaluation of nano-micelles in in vitro tumor models. We recognize that understanding the downstream signaling pathways activated by nano-micelles is crucial for gaining a deeper understanding of their mechanisms of action and optimizing their therapeutic effects.
Although our study has not directly delved into this area, based on existing literature, the mechanism of action of nano-micelles in tumor treatment may involve multiple signaling pathways. For instance, studies have shown that hyaluronic acid-modified nano-micelles can enhance the uptake of tumor cells through CD44 receptor-mediated endocytosis, which may involve changes in cellular signal transduction. Additionally, chitosan-based nano-micelles have shown potential in antitumor immunotherapy; they are capable of activating the cGAS-STING signaling pathway, which is an important DNA sensing pathway that regulates both innate and adaptive immune responses.
These research advancements indicate that nano-micelles may influence the biological behavior of tumor cells through various mechanisms, including the activation of signaling pathways. We plan to explore these potential mechanisms in future studies to more comprehensively understand the targeting mechanisms and therapeutic effects of nano-micelles.
Round 2
Reviewer 1 Report
Comments and Suggestions for Authors
I thank the authors for their detailed work in answering the questions. The authors have done an excellent job. It is with great pleasure that I recommend the publication of this paper in Molecules.